# Surrounded by Sound: The Impact of Tinnitus on Musicians

**DOI:** 10.3390/ijerph18179036

**Published:** 2021-08-27

**Authors:** Georgina Burns-O’Connell, David Stockdale, Oscar Cassidy, Victoria Knowles, Derek J. Hoare

**Affiliations:** 1British Tinnitus Association, Unit 5 Acorn Business Park, Woodseats Close, Sheffield S8 0TB, UK; david@tinnitus.org.uk (D.S.); oscarcassidymusic@gmail.com (O.C.); 10vknowles@gmail.com (V.K.); 2NIHR Nottingham Biomedical Research Centre, Ropewalk House, 113 The Ropewalk, Nottingham NG1 5DU, UK; Derek.Hoare@nottingham.ac.uk; 3Hearing Sciences, Mental Health and Clinical Neuroscience, School of Medicine, University of Nottingham, Nottingham NG7 2UH, UK

**Keywords:** tinnitus, musicians, occupational noise exposure, hearing health, lived experience, public health

## Abstract

AIM: To investigate the impact of tinnitus on professional musicians in the UK. BACKGROUND: Tinnitus is the experience of sound when an external source is absent, primarily associated with the ageing process, hearing loss, and noise exposure. Amongst populations exposed to industrial noise, noise exposure and noise-induced hearing loss (NIHL) have been found to be the factors most associated with tinnitus. The risk of NIHL amongst professional musicians is greater than that amongst the general population, meaning they may be at increased risk of tinnitus. METHODS: Seventy-four professional musicians completed an online survey involving closed and open-ended questions, and completed the Tinnitus fuctional Index (TFI) questionnaire. Descriptive statistics and thematic analysis of open-ended qualitative responses were used to analyse the data. RESULTS: Three themes were generated from the analysis of the responses to the open-ended questions. These themes were: (1) the impact of tinnitus on the lives of professional musicians, (2) professional musician experience of tinnitus services, support, and hearing health and safety, and (3) the support professional musicians want. The mean global TFI score for professional musicians was 39.05, interpreted as tinnitus being a moderate problem. Comparisons with general population data revealed lower TFI scores for the TFI subscales of ‘sense of control’ and ‘intrusiveness’ for professional musicians and higher for auditory difficulties associated with tinnitus amongst professional musicians. CONCLUSION: Tinnitus can negatively impact on professional musicians’ lives. There is a need for bespoke self-help groups, awareness raising, and education to prevent tinnitus and promote hearing health among musicians.

## 1. Introduction

Tinnitus is the perception of sound in the absence of an external source, which can be permanent or temporary [1]. There is uncertainty about how many people experience tinnitus, with estimates in general adult populations ranging from 5.1% to 42.7% [2]. Descriptions of tinnitus commonly include experiences of hissing, ringing or sizzling, and these experiences can be constant or intermittent and can be perceived in one or both ears, or localised to a source within the head [3]. Tinnitus is categorised as objective, resulting from a physiological phenomenon that may be heard by an examiner; or more commonly as subjective and only heard by the individual. Diagnosis of subjective tinnitus relies on self-report, with an emphasis on the functional impact of the condition on the individual [4]. For many people, tinnitus becomes bothersome. People living with tinnitus report problems including reduced quality of life, fears, inability to concentrate, effect on hearing ability and loss of peace [5,6]. The term ‘tinnitus disorder’ was recently introduced to differentiate those who experience both the auditory component and associated suffering, from those who experience the auditory component but no difficulties with it [7].

Causes of tinnitus are debated, but it is often viewed as a symptom or comorbidity related to other conditions such as the ageing process [8], hearing loss [9] and noise exposure [10,11,12]. Exposure to high levels of noise in the workplace has contributed to hearing loss [13]. Unsafe exposure to noise is common in some workplaces including construction, farming, and the music industry [14,15,16,17,18]. For those working in construction and the music industry, exposure to occupational noise is the biggest risk factor associated with tinnitus and hearing difficulties [18]. Professional musicians (PMs) perform and practice music on a regular basis leading them to be at high risk of noise-induced hearing loss and tinnitus [19]. 

Professional musicians have a unique set of stressors and risk factors relating to their health as part of their job. Musician populations have reported having health issues that differ from equivalent representative national populations. These include physiological difficulties such as musculoskeletal pain [20,21], psychosocial concerns [22,23] and a variety of psychological factors, including stress, depression, and anxiety [24]. 

Tinnitus can be debilitating and have a great impact on musicians private and professional lives [25]. A study by Schmidt et al. (2019) noted that the presence of tinnitus, whilst not consistently identified as attributed to noise-induced hearing loss (NIHL) may be related to noise exposure, meaning musicians can be considered a high-risk group and that the percept has potential to interfere with professional functioning, as well as outside of professional work. Schmidt et al. (2019) used a questionnaire which included two questions relating to whether musicians found tinnitus to be “a problem” during and/or outside of orchestral life. Although the breadth of the study did not explore the specific ways in which tinnitus may impact the professional and non-professional lives of musicians, it highlighted some interesting findings. One such finding was that musicians generally tended to find tinnitus more of a problem outside of work rather than within. This could be due to the masking effect of sound during orchestral work or the decreased opportunities to attend to tinnitus sensations when actively listening to musical stimuli (at work) as opposed to outside. Whilst discussing the lack of musicians actively seeking hearing help, a study by Greasley et al. (2020) theorised that musicians may identify their own solutions to cope [26], and Schneider et al. (2009) voiced the possibility that musical training may act as a protective measure against the impact of tinnitus on a musician [27]. 

When looking at non-professional musicians, Schmuziger et al. (2006) conducted an audiological evaluation and found that 17% of the sample experienced tinnitus [28]. In comparison to the PMs who participated in other studies, these participants did not present any clinical psychological distress. A comparison of distress experienced by professional and non-professional musicians might better indicate whether tinnitus presents more of a concern when the individual is reliant upon their hearing for professional purposes or whether psychological factors distinct to PM lifestyles may affect the interaction of tinnitus and mood.

It is possible there is a significant element of shame or embarrassment associated with tinnitus in PMs. Jansen et al. (2008) investigated NIHL and other hearing complaints amongst orchestral musicians and found that 19% of the sample reported that they would feel “ashamed” to have a hearing disorder [29]. Reasons given included thinking they would be a worse musician or that their colleagues might doubt their abilities as a musician. A small number (7%) even indicated they worried their job may be at risk if management became aware of any hearing disorder. Jansen et al. (2008) reported that such was the concern around hearing ability that some participants were reluctant to discuss the topic or agree to measurements associated with hearing problems [29]. As a result, available data may be subject to a comparative lack of reliability regarding epidemiological reporting. Studies employing anonymous responses alongside qualitative analysis may assist in providing reliable data regarding prevalence and degree of impact of tinnitus amongst musicians. 

Prior research has explored the impact of NIHL in musicians [30,31,32,33,34,35,36,37] the genre of music played [19,38,39], and the type of instrument played and the influence of the position of the instrument or musician on the experience of tinnitus [19,35,40,41,42]. There is still little known however about the lived experience of tinnitus for musicians. The aim of this study was to explore the impact of tinnitus on professional musicians in the UK, and to understand the support they receive or need because of tinnitus. 

## 2. Methods

This work is reported according to the Checklist for Reporting Results of Internet E-Surveys (CHERRIES) [43].

### 2.1. Design

This study involved an open survey of professional musicians, i.e., a convenience sample.

### 2.2. Ethical Review

This study was reviewed and given favourable opinion by the University of Nottingham Faculty of Medicine and Health Sciences Research Ethics Committee (ref: 471–2001). Participation in the study was entirely voluntary. As stated in the participant information at the start of the questionnaire, completion and subsequent submission of a questionnaire was taken as informed consent, therefore separate written informed consent was not required. Consideration that participants could potentially experience their tinnitus in a more bothersome way once they had participated in the research and completed the survey was stated. However, the potential for negative consequences of participation were considered low and outweighed by the potential positive experience of informing future tinnitus services and support for professional musicians.

### 2.3. Development and Pre-Testing

The questionnaire consisted of three eligibility questions, demographic information, the tinnitus functional index (TFI) [44], and 10 open-ended questions which were developed specifically for this research (Appendix A). The open-ended questions asked about general health, the impact tinnitus has on their everyday life, and what support/health care they received or would consider helpful. There was patient and public involvement throughout the research process; the questionnaire was developed with advice from professional musicians and people who have lived experience of tinnitus. Before being data collection began, the questionnaire was administered on two musicians and two people who had tinnitus to test the usability, functionality and to check the wording of the questions (excluding the TFI section) on the survey. Following this piloting of the questionnaire, minor amendments were made based on the feedback provided by those involved in the PPI consultation. These amendments included changes to the some of the questions, including corrections to spelling and improvements to the order of some open-ended questions. The musicians who had piloted the survey also helped with participant recruitment by sharing information about it with their contacts who were also musicians.

As mentioned above, in addition to the self-devised questionnaire, participants were asked to complete the TFI [45]. The TFI is a self-report, composite, questionnaire which was included in the survey instrument developed for this study. There are 25 items and eight subscales on the TFI; intrusiveness (unpleasantness, intrusiveness, persistence), sense of control (reduced sense of control), cognition (cognitive interference), sleep (sleep disturbance), auditory (auditory difficulties attributed to tinnitus), relaxation (interference with relaxation), quality of life (quality of life reduced), and emotional (emotional distress). The participant scores each item on a 10-point scale and the total score is divided by 2.5 to give a total score out of 100. The TFI shows high internal consistency (α = 0.80), high test-retest reliability (ICC = 0.86), an acceptable level of test-retest agreement (93%), and good convergent validity with other tinnitus questionnaires (>0.8) [45]. Scores less than 25 indicated mild tinnitus, scores 25–50 indicate ‘significant problems with tinnitus’ with a ‘possible need for intervention’ and scores over 50 indicate a ‘severe’ problem indicating qualification for ‘aggressive intervention’ [46].

### 2.4. Recruitment Process and Description of the Sample Having Access to the Questionnaire

Recruitment was via advertisement of the study online via social media and on musician and tinnitus online forums, with details emailed to relevant charities and organisations for dissemination. Potential participants were directed to a study specific web link. To be eligible to participate in the research, the participants were required to:earn 50% of their income from music;have experienced tinnitus;be aged over 18;reside primarily in the UK and be eligible to work in the UK.

### 2.5. Survey Administration

The survey was administered online using SurveyMonkey. A paper version of the questionnaire was available on request—none were requested. On entering the survey, participants were presented with and asked to read the Participant Information Sheet.

The questionnaire design meant that the participants were unable to skip items on the TFI section of the questionnaire, but they were able to skip the open-ended questions at the end and six participants opted to skip those questions. At the end of the survey, participants were signposted to resources available from the British Tinnitus Association (BTA) and Help Musicians in case they wanted support or information after their participation in the research.

Online survey data were password protected and stored on a secure server and was only accessible by the research team. To ensure anonymity, each musician who completed the questionnaire was assigned a unique study identity code number for their entry.

### 2.6. Response Rates

The survey was accessed by 280 people. Of those, five did not read the participant information sheet so could not continue: 30 read the participant information sheet and chose not to participate. Of the 173 who started to answer the initial screening questions 45 did not identify as a professional musician, three had not experienced tinnitus, 34 did not reside primarily in the UK, and one was under 18 years of age, so were excluded at this stage. Six further participants completed the screening questions but did not continue further. Eighty-four participants answered the demographic questions, 74 completed the TFI items, 68 completed the open-ended questions, and 41 provided additional comments. There were no missing data for the TFI global score. The total number of eligible participants whose data was included in the analysis was 74. 

### 2.7. Preventing Multiple Entries from the Same Individual

To prevent multiple entries from the same individual any duplicated IP addresses were deleted. 

### 2.8. Analysis

Questionnaires which were terminated early were not included in the analysis. Descriptive statistical analysis was used to gain an overview of the findings from the close-ended questionnaire responses. This analysis was conducted using Microsoft Excel software (Microsoft Corporation, Santa Rosa, CA, USA). Analysis of TFI scores determined which domains measured by the tool were particularly problematic for musicians. An independent-samples t-test was conducted to compare mean TFI subscale scores in professional musician and general research population samples. Data were obtained from the PM population in the present study (N = 74) and compared with equivalent mean TFI scores from a general research population sample obtained from Fackrell et al. (2016) study. Available data were recoded and inserted into Statistical Package for the Social Sciences (SPSS) software (IBM Corp: Armonk, NY, USA) [47]. Since individual data points spanning the present study and Fackrell et al. (2016) study were unobtainable, a ‘summary independent samples T-test was conducted in SPSS using the mean score, standard deviation and sample size for each TFI subscale within each population sample (see Appendix A). To account for multiple comparisons of subscales statistical significance was determined as *p* ≤ 0.00625. 

Reflexive thematic analysis was used to identify patterns of meaning and key themes within the data [48]. Reflexive thematic analysis is used to explore peoples’ experiences, views and perceptions and to identify patterns of meaning across qualitative datasets. The thematic analysis used in this study was approached in a deductive way meaning the coding and theme development were guided by three research questions which were directed by existing research and concepts relating to tinnitus and musicians. The research team (GBOC and VK) conducted thematic analysis of the data from the open-ended questions using the six-phase process set out by Braun and Clarke [49] as a guide. To start with, GBOC and VK familiarized themselves with the data (Phase i), and then started coding the answers (Phase ii) which led to initial themes being generated (Phase iii). These stages of the analysis were iterative and flexible, allowing us to revisit and review the themes (Phase iv) and the subthemes within them. Finally, suitable titles and narratives were agreed for each theme and its subthemes (Phase v), and the relevant data extracts were chosen (Phase vi) which allowed for the final phase of writing up the analysis. 

## 3. Results 

### 3.1. Participant Demographics

The questionnaire was completed by 74 participants. Of those who reported their gender, 70% (*n* = 51) identified as male and 30% (*n* = 22) as female (with one non-response). Age was reported in six categories with 27% (*n* = 20) of participants reporting being 40-49 years old making this the most common age group. This was followed by the 30-39 years and 50-59 years with 24% (*n* = 18) and 19% (*n* = 14) respectively. One participant was aged 18-19 years, whilst 14% (*n* = 10) were aged 20-29 years, 12% (*n* = 9) were aged 60 years or older, and two participants did not provide their age. When asked about their ethnicity, 66 participants (89%) identified as white UK, two participants identified as from a different white background, one participant was white Asian, another participant was Asian/British, and one participant identified as Eurasian. Participants also reported their sexual orientation with 72% (*n* = 53) identifying as heterosexual/straight and 16% (*n* = 12) of those who answered the question (*n* = 74), identified as lesbian, gay, bisexual, or queer. Most responses (93%, *n* = 63) of those who answered (*n* = 68), reported experiences/diagnoses of other conditions/illnesses as well as tinnitus. The most reported comorbidities were mental health conditions (25%, *n* = 21), and other hearing-related disorders (11%, *n* = 9).

Participants reported the range of years they had lived with tinnitus: 1 year or less (*n* = 13); 2-5 years (*n* = 31); 6-10 years (*n* = 10); 11-20 years (*n* = 11); 21-30 years (*n* = 7); and over 30 years (*n* = 2). Sixty-seven participants reported how they thought they acquired tinnitus and these responses were grouped into six categories. Over half (51%, *n* = 34) of respondents reported getting tinnitus due to ‘sudden or prolonged exposure to loud noise/music’. Other reasons for developing tinnitus reported were pre-existing conditions or other hearing-related disorders (18%, *n* = 12), personal use of specific instruments or their proximity to others (15%, *n* = 10), stress (10%, *n* = 7), and other causes such as family history (4%, *n* = 3). Nine percent of participants (*n* = 6) were unsure what had caused their tinnitus. Ineffective use or non-use of hearing protection devices (HPDs) was also reported as a cause for tinnitus (22%, *n* = 15). Participants reported on how often they wore HPDs while at work. Almost one in five participants (19%) said they always used HPDs, and 26% (*n* = 19) stated they usually wore HPDs. However, almost one-quarter (23%) of respondents said they never wore HPDs (*n* = 17).

### 3.2. Lived Experiences of Professional Musicians Who Have Tinnitus

Thematic analysis of responses to open ended questions determined three broad themes and seven subthemes (Table 1). It is important to highlight that there was natural overlap between themes, due to the complexities in the everyday lives experienced by the professional musicians who have tinnitus.

#### 3.2.1. Theme 1: The Impact of Tinnitus on the Lives of Professional Musicians

The first theme emerging from the data encompassed how tinnitus affected the life of respondents. Each participant experienced tinnitus in their own distinct ways, and different aspects of their lives were affected by it. There were two distinct subthemes which described the ‘impact of tinnitus on career’ and the ‘impact of tinnitus on personal life’. The impact tinnitus had on their work was particularly relevant, given the importance placed on listening ability within their career and how tinnitus influenced this.

##### The Impact of Tinnitus on Career

Several musicians described how tinnitus affects them at work *“especially in a job with such reliance on listening” (PM66).* For some, when they were working, their tinnitus would impact on their concentration because they rely on their hearing to produce music: 

PM2: It does get annoying and can have an impact on concentration during solo practice. It can also be incredibly frustrating in quieter moments of the music, in orchestra or chamber music for example, to start hearing tinnitus.

Furthermore, some musicians highlighted that tinnitus restricted their ability to do their job, and influenced the quality of the work they produced, as they struggled to navigate varying sound inputs:

PM42: I am sometimes unsure if I am hearing a high-pitched note from somewhere, or if it is in my head. This can affect my perception of harmony.

Tinnitus could also affect the jobs that musicians felt comfortable doing and how they felt whilst doing their work. One participant had refused to work at certain events depending on the noise exposure at the venue, while another shared how tinnitus could impact on their mood and because of this they struggled to feel positive about their work. In contrast to the negative impact experienced by musicians, a small number of musicians reported that their profession could be beneficial in managing their tinnitus: 

PM2: Fortunately, being a musician and being surrounded by sound means that I often don’t notice my tinnitus.

PM72: Normally it has no impact as I find playing music masks the tinnitus. It’s when I stop is when it starts to impact my life.

##### The Impact of Tinnitus on Personal Life

For many participants, tinnitus hindered social life and led to avoidance of certain events or settings, which in turn led to feelings of isolation. One participant specified that their tinnitus was often worsened by the levels of stress they experienced in social situations. As a result, their tinnitus could frequently dictate how they navigated social activities and events. For many individuals, tinnitus often acted as a barrier to maintaining conversations with others, disrupting their social interactions:

PM24: I get frustrated easily because I find it hard to follow conversation.

PM61: It [tinnitus] can exclude you from conversations.

In addition, the participants described their struggle with finding peace and quiet, as well as feelings of loneliness: 

PM54: I am in mourning for the loss of personal peace and quiet.

PM56: A lonely place to be in [yo]ur head when no one can hear it but you.

PM72: At times the tinnitus can make you feel very isolated, alone and extremely drained.

As indicated in the TFI scores participants found relaxation difficult due to their tinnitus. This was reflected in the open-ended question data, as individuals described problems with relaxing and/or sleep:

PM28: I never experience absolute silence, which make it difficult to relax, and/or sleep.

PM4: Struggle with watching tv, relaxing, sleeping.

PM2: It [tinnitus] impacts my ability to wind down and relax.

The isolation that living with tinnitus could cause was discussed in the context of relating to those who have no understanding of tinnitus. One participant described their head as *“a lonely place…when no one can hear it but you”* (PM50). Some participants also outlined wanting more understanding and awareness from those who do not have tinnitus, stating that *“some awareness would help with [the] stigma”* (PM30). Some participants further described how tinnitus negatively impacted on their mental health and emotional wellbeing:

PM2: The inability to experience complete silence can be quite depressing.

PM8: Anxiety affects [my] relationship with family.

PM54: [tinnitus] has been [the] cause of depression in the past.

#### 3.2.2. Theme 2: Professional Musician Experience of Tinnitus Services, Support and Hearing Health and Safety

Participants described the support and information they had received from various sources at different stages of their career. This theme contains three subthemes, which are, ‘support/awareness in education and employment’, ‘healthcare/treatment accessed’ and ‘support from family and friends’.

##### Support/Awareness in Education and Employment

Participants described the support or information they received whilst in education, as well as their experiences in the workplace. Many discussed how this advice was frequently too vague, and more of a general warning that was not very useful in educating or protecting their hearing health:

PM64: Just told to be careful. There wasn’t much in the way of actual education.

PM8: Non[e] of the commercial venues I have worked in have ever offered hearing protection as part of PPE [personal protective equipment].

In contrast, other musicians stated that they had received effective hearing health advice from specific organisations or teachers:

PM28: I have worn earplugs since I started as a professional (at 15, I’m now 30), due to a drum tutor with chronic tinnitus and other older musicians with hearing loss, who warned of the dangers.

PM51: In a [name of place] they did regular ear tests.

PM4: Our employer provided decent earplugs free of charge.

##### Accessing Healthcare

A common finding was the professional musicians had unsatisfactory experiences with healthcare. Many respondents received basic healthcare support limited to an initial diagnosis or general practitioner appointments which included “assessments, information, referrals” (PM59) and “hearing tests” (PMs 22, 35, 46, 69). Some individuals expressed their disappointment with this level of support from healthcare professionals (HCPs), finding it inadequate. Many also described their experience as negative in tone and unhelpful due to the conduct of HCPs and the dismissive assertion that there was “nothing they could do” (PM69).

In contrast to this more general support, some participants were met with more success when accessing support from organisations or individuals with a specialist knowledge of tinnitus. This included HCPs specialising in tinnitus or hearing health, as well as the charity support groups or helplines. Mental health disorders associated with tinnitus were also highlighted by individuals who reported accessing psychological support. Although some participants discussed their unhelpful experiences with healthcare or tinnitus treatment, one participant discussed how useful their cognitive behavioural therapy had been: 

PM23: I looked into mindfulness meditation and he referred me to the local mental health department. I had weekly cognitive behavioural therapy sessions with a young therapist who had previously treated patients with tinnitus. It was a slow process, but it was the start of my recovery. Mindfulness became more useful as I progressed.

Other respondents received resources such as hearing aids to help manage their tinnitus:

PM27: Hearing aids for the last 4 years—helps a lot if they’re tuned to match the tinnitus.

##### Support from Family/Friends

The importance of social support networks in coping with tinnitus was mentioned. Many respondents talked about their tinnitus with family members and reported this was helpful in lifting their mood and dealing with feelings of stress:

PM58: My wife is very supportive and will always talk about it if I need to. She gets quite upset about me having tinnitus but always wants to do whatever she can to make me feel better. My mother is also very supportive.

One respondent also stated that their family were “sympathetic” and “starting to understand” how tinnitus could affect their mood. Others described how their family and friends provided support by making adjustments because of their tinnitus and the impact it can have in their social and everyday lives:

PM60: My family and friends understand it can be frustrating for me to ask them to repeat themselves! They also know to give me space or play white noise to help me cope.

Several respondents described how this sort of understanding and empathy was beneficial to them when they received it from others who have also experienced tinnitus. They also discussed how it was “very reassuring” to share similar experiences with fellow musicians who have managed to limit the impact tinnitus had on their career:

PM8: Other local musicians/artists with tin[n]itus. Facebook groups. Hearing other people stories and realising there are many other people going through the same thing was very reassuring.

PM46: Finding out about others who suffer and still work in music.

#### 3.2.3. Theme 3: The Support Professional Musicians Want

Respondents explicitly discussed the tinnitus support or services they wanted access to in the future. This largely addressed improvements that should be made to the means of support already available, which had been accessed previously to varying extents with mixed result. Within this theme are two subthemes which described how participants wanted ‘better awareness and understanding’ of tinnitus, and improvements respondents felt would be beneficial for ‘healthcare/ professional support’.

##### Better Awareness and Understanding

Several participants reported never discussing their tinnitus with other people. For many this was largely because they felt there was a lack of understanding from other people who do not have tinnitus. They expressed frustration that some people were unable to accommodate the impact that their tinnitus could have on their everyday life, or appreciate how it affects them: 

PM66: I haven’t really discussed it with anyone else. 

PM71: I don’t think people without it have any understandable framework to appreciate how disruptive it can be.

Building on this frustration, some respondents indicated that there should be increased general awareness of tinnitus and the negative impact it can have. The sharing of general information and reassurance about tinnitus could be effective in helping musicians to gain an understanding of their experience of tinnitus:

PM12: Simply understanding tinnitus and knowing it is not the end of the world. 

Many discussed the need for preventative education in a range of contexts, as well as advice about the risks of working in their profession:

PM67: Better understanding of the risks, less expectation that musicians should simply sacrifice their health for everyone else’s enjoyment.

##### Healthcare and Professional Support

Participants described the support they want from healthcare services to manage the impact of their tinnitus on their everyday lived experiences. Many discussed how they wanted increased availability of healthcare services and resources, as well as improved access to tinnitus support. The training of HCPs was identified as a priority with one participant explaining their experience in a tinnitus appointment with an HCP was that they displayed outdated attitudes and “zero knowledge or empathy” (PM38). 

While a limited number of individuals mentioned their experiences with tinnitus-specific services, many others felt they did not have but wanted access to tailored support:

PM43: A more specialised group/ organisation that specifically deals with performers/musicians. Giving better help support and resources for them.

In addition to management of the tinnitus, several respondents wanted support that would address the psychological effect that tinnitus had on their mental health, stating the need for psychological therapies and support:

PM14: Counselling. If tinnitus is currently irreversible then the need is to cope with living [with tinnitus] and how to avoid thinking about it. 

PM12: Mental health support is key.

To help ensure there was better awareness and understanding of tinnitus amongst music professionals and across the industry, respondents felt future work could look to places of music education to implement information-sharing interventions. These could encourage preventative behaviours, reduce the experience of tinnitus amongst musicians, and therefore decrease the need for tinnitus support later in the professional musician’s career:

PM1: I think more needs to be done in UK conservatories for classical musicians. The minute you enter the junior schools, undergraduate and postgraduate programmes, [music] students should have compulsory education on hearing loss, tinnitus and have compulsory hearing tests.

PM6: More education for people at the start of careers. More visible signposting in music venues and nightclubs to warn people. More legislation/enforcement of legislation around exposure to loud noise in work environments.

### 3.3. Tinnitus Impact 

The severity of tinnitus as measured using the TFI ranged from zero to 99 across the sample, with a mean overall TFI score of 39.05 (SD = 22.12) out of a possible 100. Table 2 shows the average participant scores on each TFI subscales, in order of highest to lowest mean score. The three highest mean scores were for the effect of tinnitus on ‘relaxation’, ‘sense of control’, and for tinnitus ‘intrusiveness’. ‘Relaxation’ was the only subscale which was scored as >50 suggesting a ‘severe problem’ which could meet the requirements for ‘aggressive intervention’. All other subscale scores indicated a ‘significant problem’ with a ‘possible need for intervention’ [46]. 

An independent-samples t-test was conducted to compare mean TFI scores in professional musician and general research population samples (Table 3). After adjusting for multiple comparisons only the difference in intrusiveness score was statistically significant at *p* = 0.004 in that intrusiveness was lower in PMs than the general population. 

## 4. Discussion 

### 4.1. Impact of Tinnitus on Professional Musicians in the UK

This study explored the impact of tinnitus on the lived experience of professional musicians in the UK. The data collected via the mixed-methods approach found that tinnitus affected the lives of PMs and highlighted the heterogeneity in the experiences of tinnitus within the diverse lifestyles of PMs. The TFI scores showed that tinnitus was a ‘significant problem’ with a ‘possible need for intervention’ [46] for PMs. Whilst the PMs TFI scores were generally lower than observed in a general tinnitus research population, previous work with musicians has highlighted the potential for feelings of shame and stigma surrounding tinnitus as well as fear for the security of their employment due to their hearing difficulties [29]. This potential methodological barrier in researching this topic presents the need for further work to explore whether there is an interaction between the stigma associated with tinnitus for musicians and whether this influences the way they report their experiences of living with tinnitus. 

It has been reported that musicians tend to seek clinical help for tinnitus less than non-musicians [26]. This could be due to musicians developing their own coping strategies based on skills unavailable to non-musicians, it could also be explained by musicians’ fear of negative associations with the condition [29], or it could be because of previous negative experiences. Responses to the open-ended questions highlighted a lack of empathy in the support and understanding provided to PMs from different sources. This includes the ‘stigma’ and negativity surrounding tinnitus among the general public, as well as among family and friends, and healthcare professionals. 

Similarly to experiences of military veterans who have tinnitus [50], a lack of knowledge and understanding was also an identified as an issue in healthcare settings with some participants saying HCPs were unhelpful during appointments. Training events and signposting to accurate and up to date information could help to educate and inform those who live or work with musicians about the impact that tinnitus can have on their everyday lives.

When considering the TFI subscale scores, the PMs scored highest on the ‘relaxation’, ‘sense of control’, and ‘intrusiveness’ subscales. Participants’ tinnitus had impacted on the musicians’ experiences of relaxation. This was the only mean TFI subscale score above 50 indicating a ‘severe problem’ which could qualify for ‘aggressive intervention’ [46]. The data collected from the open-ended questions supported these findings, evidencing PMs’ challenges with relaxation and their inability to “enjoy peace and quiet” or “quiet resting activities”. These findings are likely to be a consequence of a lack of knowledge about tinnitus and how it can be managed. Workshop sessions and resources giving attention to and providing information about relaxation techniques and how to manage tinnitus could be helpful for PMs.

‘Sense of control’ scores, although lower than observed in a general tinnitus research population were still rather high at 46.7 points. As previous research reported, musicians are at risk of noise exposure at work [14,15,16,17,18], and they may be aware that their tinnitus has arisen from occupational noise exposure leading to some feelings of personal responsibility and control. An internal locus of control has been suggested to be a mediating factor in how concerned musicians are about their tinnitus [51] and as such may be connected to anxiety experienced. 

Intrusiveness scores, although third highest scoring subscale for PMs, was scored significantly lower than in a general tinnitus research population. This may be explained by the musicians’ work environment where they are “surrounded by sound” (PM2). This could offer a tinnitus masking effect or an auditory distraction while PMs are at work. This is consistent with reports that musicians can find their tinnitus to be more of a concern outside of their musical work environment than within it [25].

That TFI intrusiveness scores were statistically significantly lower in PMs than was observed in a general tinnitus research population is striking for several reasons. Notably, a recent consensus-based exercise involved patients, clinicians, and other key stakeholders to determine core outcomes for tinnitus trials identified intrusiveness as broadly important [52]. The exercise identified three core outcome sets that were considered critical to measure in all tinnitus trials involving sound, psychology, or drug-based treatments, and whilst each core outcome set contained multiple outcomes, only intrusiveness was common to all three. Its general importance raises the question why in PMs intrusiveness is so much less of an issue. One obvious explanation is their extensive exposure to sound and music. Music forms an important part of many sound-based or combination treatment approaches to tinnitus, which are generally shown to be equivalent to counselling or other treatment approaches [53,54,55,56]. Proposed mechanisms of music-based treatment approaches range from neural plastic change in the auditory system, relaxation, enjoyment, and attention shifting. These mechanisms therefore could be daily consequences of at least some PMs and proportionately reduce intrusiveness. It would be very interesting to compare effects of intervention between PMs and non-musicians to explore these mechanisms further. 

The ‘auditory’ subscale of the TFI is also worth mentioning as for some PMs it was scored very highly indicating it to be a main problem they experience because of tinnitus. This may be explained by the nature of the professional musicians’ career and the substantial reliance on listening ability to perform occupational duties, furthermore cognitive abilities and perceptual training are inherent in the development of musical ability and may be influential in noticing the percept more readily and accurately. For example, non-musicians would arguably be unlikely to complain that the tinnitus percept, *“can affect my perception of harmony”* (PM42). Simply put, there are more opportunities for PMs to interact with their tinnitus due to the nature of their work. In contrast to this negative impact, it has been suggested that PMs’ training and unique skillset could be an advantage in helping them identify their own ways of managing [26] and could even act as a protective factor against the impact of tinnitus [27].

### 4.2. Support PMs Receive or Need Because of Tinnitus

The impact of tinnitus was experienced by PMs in their professional and private lives, with the former being particularly problematic given that listening is an essential skill for musicians. The experience of receiving support and information had been mixed, with a minority of the PMs acknowledging that some sources of support and/or information had positive results. As a result of negative experiences where tinnitus support had been previously accessed, potentially helpful support services were identified by the PMs. 

The subjective nature of tinnitus and the negative effect it had on musicians’ social lives and interactions had led to some PMs avoiding certain social situations. This had resulted in feelings of isolation and loneliness which supports findings from previous research [57,58,59]. While tinnitus often hindered the PMs experiences of socialising and interactions with other people, communicating with other musicians who also experienced tinnitus was deemed to be beneficial. Hearing positive stories from others in the industry who had learnt ways to successfully manage their tinnitus without it hindering their career in music helped PMs to feel reassured and less isolated. Having a formalised peer-led support option for tinnitus management could help PMs to feel less isolated. In a study which explored the mechanisms by which tinnitus support groups can provide social connectedness from peer interaction, the most-valued features were the knowledge and information provided, the sense of belonging, and the creation and maintenance of a sense of hope towards tinnitus [60]. To provide this support, dedicated groups for PMs with tinnitus could be developed. Such groups would provide advice and information, opportunities to discuss experiences about living as a PM with tinnitus, and allow sharing of management techniques so that musicians can develop social connectedness with their peers and can learn about their coping strategies.

From the questionnaire results, it is apparent that the musicians would benefit from support with their tinnitus and mental health. Tinnitus is associated with mental health disorders such as anxiety or depression [24,61,62] and some studies have explored the relationship between musicians and psychological conditions associated with distress (for example stress, depression, and anxiety) and tinnitus [22,23,63]. Findings amongst non-musicians have suggested weak to moderate association between tinnitus and stress [64,65] and some evidence has suggested a bidirectional relationship between stress and self-reported hearing problems including tinnitus [66]. Furthermore, musicians experiencing tinnitus have reported experiencing more depressive symptoms than non-musicians with tinnitus [51]. Some musicians in the present study attributed their mental health problems to their experience of tinnitus and highlighted that *“mental health support is key”* (PM12). The data from this study regarding musicians’ experiences of tinnitus and mental health is supported by previous research, and in addition the desire for (improved) mental health support is highlighted.

Across the findings it was evident that this population of musicians have experiences with tinnitus that are uniquely related to their profession. For example, many identified issues surrounding HPDs, which includes finding appropriate methods of use and how to access them. The information they received during their (music) education was frequently inadequate, with some participants highlighting the lack of warning they were given surrounding noise exposure and hearing protection advice. As the data provided in this research about the musicians’ experience of education is retrospective, and therefore, reliant on memory, it is important to evaluate the current situation in tinnitus prevention and information education. If education about tinnitus is still poor or absent for young musicians, it is recommended that an education intervention is developed and implemented. This could be in the form of a module to educate young musicians about how to protect their hearing and reduce the risk of developing tinnitus. This module could be implemented in music colleges to share advice surrounding relevant issues, including using HPDs, where to access them, and how to use them effectively.

### 4.3. Limitations of the Current Study

The mixed-methods survey approach was a suitable method for gaining an overview of the challenges experienced by PMs, however some additional qualitative research would have provided more nuanced and in-depth insights and understanding of the complexities of tinnitus and the impact it can have on professional musicians. Future research should include semi-structured interviews and/or focus groups. This would enable the participants to provide more detailed descriptions about their experiences of living with tinnitus and allow the research to interrogate their experiences using follow-up questions to gain more in-depth insights [67]. This was beyond the scope of the survey design used in this study. This would provide the opportunity for more holistic understandings of the contexts of the musicians’ experiences to be documented and allow the musicians’ perspectives to be further interrogated providing a better understanding of the challenges this unique population encounter.

## 5. Conclusions

Tinnitus can be very challenging for professional musicians and while there are some differences in TFI scores between the general research population and musician population, such scores combined with the open-ended question data indicate that tinnitus is a “significant problem” for PMs. Further exploration would help to interrogate and develop enhanced understanding relating to the complexities and challenges of PMs who live with tinnitus experience. These findings provide evidence of need for interventions including dedicated peer-support groups to help reduce feelings of isolation, awareness raising and information sharing workshops to help PMs learn about how to self-manage tinnitus, and education to prevent tinnitus and promote healthy hearing behaviours among musicians.

## Figures and Tables

**Table 1 ijerph-18-09036-t001:** Themes reflecting the lived experience of professional musicians who have tinnitus.

Theme	Subthemes
1. The impact of tinnitus on the lives of professional musicians	The impact of tinnitus on career The impact of tinnitus on personal life
2. Professional musician experience of tinnitus services, support and hearing health and safety	Support/awareness in education and employment Accessing healthcare Support from family/friends
3. The support professional musicians want	Better awareness and understanding Healthcare and professional support

**Table 2 ijerph-18-09036-t002:** Tinnitus functional index (TFI) mean total score and mean subscale scores.

TFI Subscales	Mean/100	SD
TFI total score	39.05	22.12
Relaxation	50.86	31.83
Sense of Control	47.66	25.25
Intrusiveness	42.84	23.73
Auditory	40.45	29.75
Sleep	35.59	31.16
Cognitive	34.59	26.29
Emotional	34.01	27.33
Quality of Life	29.59	25.56

**Table 3 ijerph-18-09036-t003:** Tinnitus functional index (TFI) mean total scores and mean subscale scores of UK professional musicians and a UK research volunteer population.

TFI Domain	UK PMs ^a^	General Research Population ^b^	Statistical Comparison (Uncorrected)
Total TFI score (0–99)	39.05 (22.1)	40.6 (20.1)	t(355) = 0.58, *p* = 564
Relaxation	50.9 (31.8)	54.6 (29.2)	t(355) = 0.96, *p* = 0.336
Sense of Control	47.7 (25.3)	53.9 (23.2)	t(355) = 2.02, *p* = 0.044
Intrusiveness	42.8 (23.7)	52.8 (21.1)	t(355) = 3.52, *p* = 0.0005 *
Auditory	40.5 (29.8)	34.0 (27.3)	t(355) = −1.78, *p* = 0.077
Sleep	35.6 (31.2)	39.6 (32.3)	t(355) = −0.96, *p* = 0.339
Cognitive	34.6 (26.3)	35.8 (27.1)	t(355) = 0.34, *p* = 0.731
Emotional	34.0 (27.3)	30.3 (26.3)	t(355) = −1.07, *p* = 0.284
Quality of Life	29.6 (25.6)	28.2 (25.4)	t(355) = −0.42, *p* = 0.676

^a^. Present study; sample size *n* = 74. ^b^. Fackrell et al. [45]; general UK research volunteer population sample size *n* = 283. * Indicates statistical significance after correction for multiple comparisons.

## Data Availability

Data is available on request from the corresponding author.

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
