# Peer review of "Surrounded by Sound: The Impact of Tinnitus on Musicians"

_ijerph, 2021, doi:10.3390/ijerph18179036_

Round 1

Reviewer 1 Report

This manuscript highlights the impact of tinnitus on professional musicians and offers great insight into the support these professional musicians need or receive for tinnitus. This manuscript is novel in its approach, and I am pleased with the way the authors have explored complicated areas associated with tinnitus perception through the open-ended questions. This manuscript also reveals the limitations of the current health care system in dealing with patients with tinnitus. At the same time, it also suggests possible solutions which will be useful for the health care professionals’ community.  In my opinion, the paper is well written, well structured, and the sample size is adequate. This manuscript should be accepted after minor revisions.

Comments:

  • Page 2, paragraph 3, line 62 – “ Their study noted that…..” This sentence needs to be rephrased because it is not clear. I would recommend adding the name of the author of the study and rephrase the sentence.
  • Page 3, first paragraph under Development and pretesting section, line 132 – “the questionnaire was tested by two musicians …..”. Consider changing this sentence to “the questionnaire was administered on two musicians…”
  • Page 4, second paragraph, line 162-163- consider rephrasing this sentence.
  • Page 4, line 190-193 - replace “Fackrell et al. 45” with “ Fackrell et al. (2016) study”
  • Page 5, Participants demographics section- The number of participants of different age groups is not adding up to 74. (20+18+14+1+10+9= 72). Please cross-check and correct the numbers.
  • Page 13, line 592 – consider adding “of” after “challenges”.

Reviewer 2 Report

This is an interesting and timely paper. Even if the findings are not very impressive, it provides some interesting new insights. As a whole, it is well written; the language use is OK and the methodology is sound. The paper provides also an interesting mix of qualitative and quantitative research, and the readability and understandability is quite good. Care should be taken, however, to explain all used abbreviations at first appearance, and some concepts should be explained better. This holds in particular for the concept of “reflexive thematic analysis”, which has a central place in the whole paper. I recommend acceptance of the paper, on condition that all remarks and comments are addressed appropriately. I list some of them below.

Detailed comments

  • page 2, line 63: explain the abbreviation NIHL (noise-induced hearing loss) at first appearance
  • page 3, line 99: here NIHL is explained at first; this should be done before
  • page 3, line 128: explain better the patient and public involvement; not clear
  • page 3, last paragraph: please provide some more information about the TFI measurements: how were they measured? what is the meaning of borderline test-retest agreement? Is it possible also to refer more in detail to some of the subscales (simply summing them up, as in table 2) so as to be clearer?
  • page 5, line 198: please explain somewhat more in detail the concept of “reflective thematic analysis”
  • page 7, line 319: what does the abbreviation PPE mean?
  • page 7, line 330: what does the abbreviation GP mean?
  • page 7, line 334: what does the abbreviation HCP mean?
  • page 8, line 342: what does the abbreviation CBT mean? (cognitive behavioural therapy?)
  • page 10, line 439: What does UK PMsa mean? (United Kingdom Professional Musicians?) what does the “a” stand for? Is this the “a” of the table caption below? In that case a superscript could be better.
  • page 10, line 440: Where is the “b” of the table caption to be found in the table?
